# The Effects of Sevoflurane and Aβ Interaction on CA1 Dendritic Spine Dynamics and MEGF10-Related Astrocytic Synapse Engulfment

**DOI:** 10.3390/ijms25137393

**Published:** 2024-07-05

**Authors:** Qinfang Shi, Xingxing Wang, Arpit Kumar Pradhan, Thomas Fenzl, Gerhard Rammes

**Affiliations:** 1Department of Anesthesiology and Intensive Care Medicine, School of Medicine and Health, Klinikum Rechts der Isar, Technical University of Munich, 81675 Munich, Germany; sqf0526@163.com (Q.S.); Arpitkp@umich.edu (A.K.P.); thomas.fenzl@tum.de (T.F.); 2Department of Anesthesiology, Union Hospital, Tongji Medical College, Huazhong University of Science and Technology, Wuhan 430022, China; 3School of Integrative Medicine, Shanghai University of Traditional Chinese Medicine, Shanghai 201203, China; xingxingwang@shutcm.edu.cn; 4Graduate School of Systemic Neuroscience, Ludwig Maximilian University of Munich, 82152 Munich, Germany

**Keywords:** anesthesia, Alzheimer’s disease, spine dynamics, synaptic plasticity, astrocyte-mediated synaptic phagocytosis

## Abstract

General anesthetics may accelerate the neuropathological changes related to Alzheimer’s disease (AD), of which amyloid beta (Aβ)-induced toxicity is one of the main causes. However, the interaction of general anesthetics with different Aβ-isoforms remains unclear. In this study, we investigated the effects of sevoflurane (0.4 and 1.2 maximal alveolar concentration (MAC)) on four Aβ species-induced changes on dendritic spine density (DSD) in hippocampal brain slices of Thy1-eGFP mice and multiple epidermal growth factor-like domains 10 (MEGF10)-related astrocyte-mediated synaptic engulfment in hippocampal brain slices of C57BL/6 mice. We found that both sevoflurane and Aβ downregulated CA1-dendritic spines. Moreover, compared with either sevoflurane or Aβ alone, pre-treatment with Aβ isoforms followed by sevoflurane application in general further enhanced spine loss. This enhancement was related to MEGF10-related astrocyte-dependent synaptic engulfment, only in AβpE3 + 1.2 MAC sevoflurane and 3NTyrAβ + 1.2 MAC sevoflurane condition. In addition, removal of sevoflurane alleviated spine loss in Aβ + sevoflurane. In summary, these results suggest that both synapses and astrocytes are sensitive targets for sevoflurane; in the presence of 3NTyrAβ, 1.2 MAC sevoflurane alleviated astrocyte-mediated synaptic engulfment and exerted a lasting effect on dendritic spine remodeling.

## 1. Introduction

Alzheimer’s disease (AD) is an age-related neurodegenerative disease characterized by progressive cognitive decline and memory impairment, which accounts for the most common cause of dementia and is one of the greatest public health problems worldwide [1]. The neuropathology of AD, such as amyloid plaques and neurofibrillary tangles, begins 15 to 20 years prior to the onset of distinct cognitive impairments. AD patients progress from a normal cognitive state to subtle changes in the preclinical phase, then to significant brain dysfunction symptoms in the prodromal phase, and eventually to dementia and loss of the ability to perform daily activities. It is currently believed that the earliest detectable pathological change in AD is the accumulation of amyloid beta (Aβ) in the preclinical phase. In the prodromal phase, cognitive decline may occur due to increasing Aβ accumulation, tau formation, and/or initial neurodegeneration. In the dementia phase, cognitive decline is strongly associated with significant Aβ accumulation, elevated tau biomarkers, and notable neurodegeneration characterized by neuronal loss. It is increasingly recognized that the underlying pathophysiology of AD and its clinical symptoms are a continuum [2,3]. With an ever-increasing life expectancy, more people are suffering from AD. Consequently, the number of patients with AD requiring anesthesia under surgery is steadily increasing [4]. However, it is suggested that general anesthetics (especially inhalation anesthetics) may trigger or accelerate the neuropathological changes related to AD [5,6]. Moreover, it is well established that patients with AD are at high risk of postoperative neurocognitive impairments, and even mild preoperative impairments of cognition may lead to postoperative delirium [7,8]. Nevertheless, the interaction of general anesthetics with AD remains unclear. Consequently, understanding such relationships is important for performing clinical anesthesia in patients with AD.

Sevoflurane, one of the most used inhalation anesthetics in clinics, is an ether compound with fluoromethyl and 1,1,1,3,3,3-hexafluoroisopropyl as the two alkyl groups [9]. The blood–gas partition coefficient of sevoflurane is 0.69, which provides sevoflurane with the properties of rapid induction of, and fast recovery from, anesthesia [10]. In addition, due to minimal effects on the cardiovascular system and virtually no organ toxicity, sevoflurane is one of the most popular anesthetics in clinical anesthesia for elderly patients. Over years of clinical application, the safety and effectiveness of sevoflurane have been fully demonstrated. However, recently, a series of preclinical and clinical studies have suggested that sevoflurane may have potential neurotoxic effects on the AD brain, thereby accelerating the pathogenesis of AD and cognitive decline [11,12,13]. Exposure to sevoflurane has been shown to widely affect the processing of amyloid precursor protein (APP) and apoptosis, leading to increased production and aggregation of Aβ in human neuroglioma cells [14]. In mice, exposure to sevoflurane leads to increased activation of caspases, β-secretase (BACE), and Aβ aggregates in the brain [12]. In addition, exposure to sevoflurane results in a dose-dependent hyperphosphorylation of tau in the hippocampus of rodents, which persists after the exposure has ended [15]. This persistent hyperphosphorylation of tau is associated with impairments in spatial learning and memory, suggesting a potential mechanism for postoperative cognitive impairment [16]. On the other hand, studies have also shown that sevoflurane has neuroprotective effects [17,18]. Therefore, this might suggest that sevoflurane can mediate both neurotoxic and neuroprotective effects in a concentration-dependent manner as well as based on the pathological state.

Although the exact pathophysiology of AD is not yet clear, the best-known hypothesis is the amyloid cascade hypothesis [19]. The cornerstone of the hypothesis is that Aβ peptide deposition is an early causative factor of AD, which initiates a pathological cascade response leading to cognitive decline [20,21]. Aβ is a 37 to 43 amino acids peptide produced from amyloid precursor protein (APP) by proteolytic cleavage via amyloidogenic pathway [22]. In AD patients, Aβ_1-40_ and Aβ_1-42_ are the predominant isoforms, accounting for ~80–90% and ~5–10% of total Aβ [23]. Moreover, post-translational modifications of Aβ (such as oxidation, phosphorylation, and nitration) give rise to Aβ isoforms with different physiological and pathological properties [24]. AβpE3 accounts for ~25% of total modified Aβ. Another modified isoform is 3NTyrAβ. 3NTyrAβ is generated by peroxynitrite nitration at tyrosine 10 of Aβ_1-42_ [25]. Previous work have demonstrated the abolishment of CA1 long-term potentiation (LTP) by the above-mentioned four Aβ species [23,25,26,27,28,29].

Synapses are functional units of the central nervous system (CNS) that play a key role in memory and learning [30,31]. Dendritic spines are tiny membranous protrusions on the dendritic shaft, serving as the anatomical sites of most excitatory synapses [32]. Synaptic dysfunction (e.g., synaptic deficits and structural damage to synapses) precedes the onset of cognitive decline and is the primary neurobiological alteration in AD [33,34]. In addition, it has been shown that inhalation anesthetics are involved in destabilizing dendritic spine dynamics, and notably, different types of spines have different sensitivities to general anesthetics [35].

Astrocytes are the most abundant glial cells in the mammalian brain. In the CNS, a single astrocyte extends thousands of fine membranous processes, which closely interact with pre- and post-synaptic structures to form a “tripartite synapse” [36]. As such, astrocytes are closely involved in primary synaptic activities, including synapse formation, maturation, and elimination, thereby maintaining synaptic homeostasis [37,38]. In the adult hippocampus, synapses are constantly undergoing rapid formation and elimination. Astrocyte-mediated synaptic engulfment is one of the major mechanisms of synaptic elimination, and its dysfunction leads to dysregulation of synaptic plasticity and cognitive deficits in a variety of CNS disorders such as AD [38,39].

Lack of multiple epidermal growth factor-like domains 10 (MEGF10), the astrocyte-specific phagocytosis receptor, decreases the relative phagocytic capacity of astrocytes by 50% [40]. Furthermore, astrocytes are also potential targets of anesthetics, and growing evidence suggests that astrocyte malfunction is involved in postoperative neurocognitive deficits [41,42]. Previous studies have found that aged mice administered with clinical doses of etomidate developed long-term cognitive deficits closely related to synaptic inhibition induced by hippocampal astrocyte dysfunction [41].

It is proposed that exposure to general anesthetics may trigger long-term morphological and functional changes in the brain [43]. However, preclinical studies suggest that anesthetics produce both neurotoxic and neuroprotective effects [44,45]. Specifically, at low concentrations for a short duration, general anesthetics are sublethal stress factors and induce endogenous neuroprotective mechanisms, whereas, at high concentrations for a prolonged duration, general anesthetics become lethal stress factors that induce neurotoxicity. Accordingly, to explore the interaction of sevoflurane with Aβ-dependent pathophysiology, we investigated the effects of two concentrations (a lower maximal alveolar concentration (MAC) of 0.4 and a higher MAC of 1.2) of sevoflurane on dendritic spine density (DSD), astrocyte-mediated synaptic engulfment, and the effect on the expression of astrocyte-specific phagocytic receptors MEGF10 in the presence of Aβ species.

## 2. Results

### 2.1. Effects of Sevoflurane and Its Interaction with Aβ Isoforms on DSD

#### 2.1.1. Sevoflurane at 0.4/1.2 MAC Decreases DSD

To clarify the effects of clinic-related concentration of sevoflurane on CA1 dendritic spine dynamics, we incubated hippocampal slices with 0.4/1.2 MAC sevoflurane. We found that sevoflurane decreased DSD of acute hippocampal slices at both concentrations (Figure 1C,E, control vs. 0.4 MAC Sevo: *p* = 0.0809; control vs. 1.2 MAC Sevo: *p* = 0.0003, Dunnett’s test of one-way ANOVA). Studies on sevoflurane’s effects on the subtypes of dendritic spines are rare. Here, we further evaluated the specific spine type density. The thin spines were reduced in under 0.4 and 1.2 MAC sevoflurane (Figure 1C,F, control vs. 0.4 MAC Sevo: *p* = 0.0884; control vs. 1.2 MAC Sevo: *p* = 0.0042, Dunnett’s test of one-way ANOVA). Moreover, 1.2 MAC sevoflurane decreased the density of stubby spines (Figure 1C,G, control vs. 1.2 MAC Sevo: *p* = 0.0270, Dunnett’s test of one-way ANOVA). In comparison, neither 0.4 nor 1.2 MAC sevoflurane decreased the density of mushroom spines (Figure 1C,H).

After the removal of sevoflurane, i.e., washout for 90 min, in hippocampal slices, the effect of 0.4 MAC sevoflurane on DSD was reversed, whereas for 1.2 MAC sevoflurane, a residual effect on total DSD was observed (Figure 1D,I–L, control vs. 0.4 MAC Sevo washout: *p* = 0.8362; control vs. 1.2 MAC Sevo washout: *p* = 0.0029, Dunnett’s test of one-way ANOVA). This is likely manifested by the persistent reduction of thin-type spines after a 90 min washout (Figure 1D,J, control vs. 0.4 MAC Sevo washout: *p* > 0.9999; control vs. 1.2 MAC Sevo washout: *p* = 0.0013, Dunnett’s test of one-way ANOVA).

#### 2.1.2. Effects of Aβ Pre-Treatment 90 min + Sevoflurane 90 min on DSD

A previous study well demonstrated that different Aβ species, i.e., Aβ_1-40_, Aβ_1-42_, AβpE3, and 3NTyrAβ, concentration-dependently inhibit CA1-LTP [29]. Here, we examined the effects of sevoflurane on Aβ-induced changes to DSD (Figure 2 shows the representative dendritic spine images). To clarify this, we pre-incubated the acute hippocampal slices with each Aβ isoform for 90 min, after which 0.4/1.2 MAC sevoflurane was added for 90 min. Thereafter, the total and subtypes of spine density were quantified.

For the total spines, we found that except for AβpE3, the other three Aβ isoforms decreased total DSD (Figure 3A, control vs. Aβ_1-40_ 180: *p* = 0.0132; control vs. Aβ_1-42_ 180: *p* = 0.0305; control vs. 3NTyrAβ: *p* = 0.0002; Dunnett’s test of one-way ANOVA). Moreover, total DSD was lower in the 3NTyrAβ group in comparison to the AβpE3 group (Figure 3A, AβpE3 180 vs. 3NTyrAβ 180: *p* = 0.0303, unpaired *t*-test), indicating a more toxic effect of 3NTyrAβ than AβpE3 on dendritic spines. Moreover, all Aβ isoforms decreased thin but no other types of spines significantly (Figure 3B–D, control vs. Aβ_1-40_ 180: *p* = 0.005; control vs. Aβ_1-42_ 180: *p* = 0.009; control vs. AβpE3 180: *p* = 0.0074; control vs. 3NTyrAβ: *p* = 0.0004; Dunnett’s test of one-way ANOVA).

A similar effect on total DSD was found between Aβ_1-40_ and Aβ_1-42_, as the total dendritic spines were reduced in the presence of Aβ_1-40_/Aβ_1-42_ (Figure 3A, control vs. Aβ_1-40_ 180, *p* = 0.0132; Figure 3A, control vs. Aβ_1-42_ 180, *p* = 0.0305), and additional 0.4/1.2 MAC sevoflurane promoted the downregulation of DSD (Figure 4A, Aβ_1-40_ vs. Aβ_1-40_ + 0.4 MAC Sevo: *p* = 0.0258; Aβ_1-40_ vs. Aβ_1-40_ + 1.2 MAC Sevo: *p* = 0.0304; Figure 4B, control vs. Aβ_1-42_ + 0.4 MAC Sevo: *p* = 0.0488; control vs. Aβ_1-42_ + 1.2 MAC Sevo: *p* = 0.0094). This combination detrimental effect was also observed in 3NTyrAβ + 0.4 MAC Sevo, but not 3NTyrAβ + 1.2 MAC Sevo, in comparison with 3NTyrAβ (Figure 4D, *p* = 0.0446), indicating a protective effect of sevoflurane under 3NTyrAβ pretreatment. Although AβpE3 did not alter DSD, the additional treatment with 0.4/1.2 MAC sevoflurane reduced the total spine density (Figure 4C, control vs. AβpE3: *p* = 0.0187/0.0151).

We further analyzed the dynamics of dendritic spine subtypes among all conditions (Table 1, with detailed statistics). CA1-thin spines were decreased among all groups of Aβ + Sevo compared with the control group. Compared with Aβ180 min groups, an increase in thin spines was found only in the 3NTyrAβ + 1.2 MAC Sevo group, indicative of a partial recovery of thin spines after application of sevoflurane into 3NTyrAβ pretreated slices.

For stubby spines, the result was more complex. Aβ_1-40_/Aβ_1-42_ + Sevo reduced stubby spines compared to either the control or Aβ group. AβpE3 + Sevo showed no effects on stubby spines, compared to control groups. However, AβpE3 + Sevo showed a reduction in stubby spines compared to AβpE3, suggesting a combination of effects of AβpE3 and sevoflurane on stubby spines. The combination effect was also observed in the 3NTyrAβ + 1.2 MAC group, as a further downregulation was detected compared with the 3NTyrAβ group. In addition, reduced stubby spines occurred in both 3NTyrAβ + 0.4 and + 1.2 MAC sevoflurane groups.

For mushroom spines, all four Aβ isoforms + 1.2 but not 0.4 MAC sevoflurane decreased the density in comparison with either the control or the corresponding Aβ, suggesting a pronounced decrease in mushroom spines by a higher concentration of sevoflurane exposure.

#### 2.1.3. Effects of Sevoflurane Removal for 90 min on DSD

To mimic an AD patient recovering from sevoflurane anesthesia, we removed sevoflurane and quantified the CA1 DSD (Figure 5 shows the representative dendritic spine images).

In accompaniment with sevoflurane washout conditions, we also checked Aβ incubation of 270 min for statistical comparison (Figure 6).

We first quantified the total DSD of all groups and found that all Aβ + Sevo washout conditions downregulated CA1 total DSD (*p* = 0.0061). Only 1.2 MAC sevoflurane produced a mild reversible effect on dendritic spine remodeling after washout in the presence of 3NTyrAβ (Figure 7D, 3NTyrAβ vs. 3NTyrAβ + 1.2 MAC Sevo washout: *p* = 0.0021, Dunnett’s test of one-way ANOVA).

We further quantified the subtypes of dendritic spines (Table 2, with detailed statistics). The thin spine density under all conditions was reduced, except the 3NTyrAβ + 1.2 MAC Sevo washout group. We saw a reversible effect of thin spines in this group, as thin spines recovered to a normal state after the removal of sevoflurane for 90 min, indicative of a positive effect of 1.2 MAC sevoflurane under 3NTyrAβ. The positive regulation of stubby spines was also found in AβpE3 + 1.2 MAC Sevo washout but not in other conditions. For mushroom spines, a reduction was detected in groups of Aβ_1-40_ + 0.4 MAC Sevo washout but not in other conditions containing 0.4 MAC sevoflurane. In comparison, the other three Aβ isoforms + 1.2 MAC Sevo washout decreased mushroom spines, suggesting a different modification mechanism on mushroom spine.

### 2.2. Effects of Sevoflurane and Its Interaction with Aβ on Astrocyte-Mediated Synaptic Engulfment

#### 2.2.1. Effects of Aβ Pre-Treatment 90 min + Sevoflurane 90 min on Astrocyte-Mediated Synaptic Engulfment

The abnormal CA1 dendritic spine dynamics may be due to astrocyte-mediated synaptic phagocytosis. Therefore, we measured the effects of sevoflurane on astrocyte-mediated synaptic phagocytosis by measuring the phagocytic index (PI) in CA1. The relative phagocytosis ability was shown as the normalized PI of the treatment group to that of the control group [40]. We found that neither 0.4/1.2 MAC sevoflurane nor 0.4/1.2 MAC sevoflurane washout affected astrocyte-mediated synaptic engulfment (Figure 8A–C,G).

#### 2.2.2. 1.2 MAC Sevoflurane Decreases Astrocyte-Mediated Synaptic Engulfment in Acute Hippocampal Brain Slices Pretreated with AβpE3/3NTyrAβ

Next, we evaluated astrocyte-mediated synaptic phagocytosis in groups of Aβ + sevoflurane. Compared to the control, 1.2 MAC sevoflurane exposure to acute hippocampal brain slices pretreated with either AβpE3 or 3NTyrAβ reduced astrocyte-mediated synaptic engulfment (Figure 9, control vs. AβpE3 + 1.2 MAC Sevo, *p* = 0.0088, one-way ANOVA followed by Dunnett’s multiple comparisons for control vs. 3NTyrAβ + 1.2 MAC Sevo, *p* = 0.0106, unpaired *t*-test).

#### 2.2.3. Removal of 1.2 MAC Sevoflurane Decreases Astrocyte-Mediated Synaptic Engulfment in Acute Hippocampal Brain Slices Pretreated Only with 3NTyrAβ

Next, we further quantified the PI values when sevoflurane was removed for 90 min (Figure 10). Surprisingly, among all groups, only 3NTyrAβ + 1.2 MAC Sevo washout for 90 min reduced the PI value (*p* = 0.0506, Figure 10E), indicating a modified synaptic engulfment under 3NTyrAβ.

### 2.3. Effects of Sevoflurane and Its Interaction with 3NTyrAβ on the Expression of MEGF10

#### 2.3.1. Neither 0.4 MAC nor 1.2 MAC Sevoflurane Affects the MEGF10 Expressions

MEGF10, an astrocyte-specific phagocytosis receptor, is essential in astrocyte-mediated synaptic elimination. We found that neither 0.4 MAC nor 1.2 MAC sevoflurane had a significant effect on the expression of MEGF10 (Figure 11).

#### 2.3.2. The Downregulation of MEGF10 Expression by Co-Application of 1.2 MAC Sevoflurane with 3NTyrAβ Is Diminished after the Removal of Sevoflurane

Xu et al. demonstrated that isoflurane displayed potential dual effects (protection or promotion) on Aβ-induced neurotoxicity depending on its concentration [46]. Here we found that 1.2 MAC sevoflurane + 3NTyrAβ downregulated MEGF10 expression in comparison with the control (Figure 12D, control vs. 3NTyrAβ + 1.2 MAC Sevo, *p* = 0.0244, unpaired *t*-test).

After removing sevoflurane, the downregulation effect on MEGF10 was diminished (Figure 13).

## 3. Discussion

With an increase in life expectancy, an ever-increasing number of AD patients need anesthesia care. However, studies suggest that inhalation anesthetics may produce neurotoxic effects and exaggerate cognitive impairment in AD patients [7,8,47]. Therefore, understanding the interaction of anesthetics with Aβ-dependent pathophysiology is important for performing anesthesia-related procedures in AD patients.

In the present study, we pre-incubated hippocampal slices with Aβ isoforms for 90 min, followed by applications of sevoflurane for 90 min, thus mimicking the clinical condition of an AD patient receiving inhalational anesthesia. We found that both sevoflurane and Aβ downregulated CA1-dendritic spines. Moreover, compared with either sevoflurane or Aβ conditions, Aβ isoforms pre-treatment followed with sevoflurane in general further enhanced spine loss. This enhancement was related to MEGF10-related-astrocyte-synaptic engulfment only in the AβpE3 + 1.2 MAC Sevo or 3NTyrAβ + 1.2 MAC Sevo condition. In addition, washout of sevoflurane alleviated the reduction of spine number in the presence of Aβ + Sevo.

Dendritic spines contribute to synaptic transmission and plasticity [32]. Therefore, an abnormal spine morphology may induce an aberrant synapse connection. Previous studies have shown that sevoflurane concentration-dependently impaired CA1-LTP in acute hippocampal slices [48], indicating an alternation of synaptic transmission under sevoflurane. The reduction in CA1 dendritic spines by sevoflurane at 0.4 MAC has been previously shown [6]. Here, we further observed a decrease in CA1-dendritic spines under 0.4/1.2 MAC Sevo sevoflurane. This may be the reason why sevoflurane concentration-dependently impaired CA1 synaptic strength in LTP.

The adverse effects of Aβ_1-40_ and Aβ_1-42_ on dendritic spines in this study are similar to previous reports. Notably, the incubation of Aβ_1-40_ and Aβ_1-42_ in acute hippocampal brain slices resulted in reduced spine density in the CA1 region [29,49]. Our present work also showed a decrease in CA1 total dendritic spines in the presence of 3NTyrAβ but not AβpE3. These different actions of Aβ species may be due to the differences in biochemical properties of Aβ isoforms. Quantifying the subtypes of CA1 dendritic spines showed a pronounced decrease only in thin spines, but not the other two spine types, among all Aβ groups, suggesting that thin spines are more sensitive than the other two spine types under Aβ treatment.

Anesthetic exposure may change Aβ properties [23]. Similarly, AD pathological state might alter with the working mechanism of anesthesia [5]. We used the Aβ pretreatment + sevoflurane combination to mimic the clinical condition of an AD patient receiving sevoflurane anesthesia. The interaction of Aβ and sevoflurane led to changes in CA1 dendritic spines. A combination of Aβ and sevoflurane generally produced a more pronounced CA1 DSD reduction compared with Aβ alone. This was mainly due to a decrease of the number of stubby spines, since in all Aβ isoforms + sevoflurane groups, stubby spines were most frequently affected as with Aβ alone. Mushroom spines seemed more sensitive to sevoflurane in Aβ + Sevo conditions, since the number of this type of spine was unchanged under Aβ + 0.4 MAC Sevo but reduced under either Aβ isoform + 1.2 MAC Sevo mixture.

The above-mentioned spine dynamics may be due to the interaction of Aβ with sevoflurane. Previous studies have demonstrated that sevoflurane can accelerate Aβ_1-40_ and Aβ_1-42_ production and, hence, induce neurotoxicity in the hippocampus [6]. This might be the reason why we observed a combined reducing effect on CA1 dendritic spines. The properties and functions of dendritic spines have been investigated in numerous studies [50,51,52]. However, the exact structure and functional differences among these spine types are still unclear [23]. Spines are classified into different types according to the size of the head and neck. A thin spine contains a small head and a long neck and is more easily changeable under stimuli compared to others, due to the shorter latency and slower decay kinetics to calcium [6]. The other two types of spines, i.e., stubby and mushroom, are more stable and mature with a larger head, especially in mushroom spines. A mushroom spine has a much larger head containing diverse organelles, allowing for more complex molecular metabolism as well as ion buffering [31]. Under Aβ + sevoflurane conditions, the different classes of spines undergo different evolutions due to their properties and interactions between Aβ and sevoflurane. However, more investigations are needed regarding the exact mechanism of how an interaction of Aβ and sevoflurane affects the different classes of dendritic spines.

To mimic an AD patient recovering from sevoflurane anesthesia, we removed sevoflurane from the mixed incubation solution while assessing dendritic spine dynamics. After sevoflurane removal, spine dynamics partially reversed to control levels e.g., a total recovery occurred for mushroom spines in Aβ_1-40_ + 1.2 MAC Sevo washout, stubby spines in AβpE3 + 1.2 MAC Sevo washout, and thin spines in 3NTyrAβ + 1.2 MAC Sevo washout. For the other groups, although an absolute dendritic spine reversal to the normal state did not occur, we observed an overall attenuation of all spine types under Aβ + Sevo washout compared with Aβ for 270 min. The total/partial recovery of CA1 dendritic spines may attenuate neural activity after Aβ + Sevo treatment. A previous study by Hofmann et al. [23] using voltage-sensitive dye imaging showed that after ~0.7 MAC sevoflurane washout, the CA1 neural activity (1) returned to normal in either Sevo washout or Aβ_1-40_ + Sevo washout or 3NTyrAβ + Sevo washout and (2) tended to fully recover in AβpE3 + Sevo washout. In line with our observations, a residual effect of sevoflurane on structure and function in CA1 after its removal is evident. This might be in correlation with delirium or cognition deficits following anesthesia, especially in patients diagnosed with AD [8].

The complex modulation of dendritic spines in the presence of either sevoflurane or Aβ alone or Aβ + sevoflurane or Aβ + sevoflurane washout may result from different mechanisms. For instance, pre- and post-synaptic elements, including voltage-dependent ion channels, ionotropic receptors and G protein-coupled receptors, contribute to synaptic transmission and spine dynamics [53,54]. Of those, the activity of astrocytes also plays a part in spine dynamics [55]. Volatile anesthetics and Aβ can trigger astrocytes to activate and assist synapse elimination through the phagocytosis process [49]. Our present study examined whether and how astrocytic synapse elimination contributes to the described changes in CA1 dendritic spines. We found that neither sevoflurane nor Aβ alone affected astrocyte-mediated synaptic phagocytosis. However, the conditions Aβ + sevoflurane or AβpE3 + 1.2 MAC Sevo or 3NTyrAβ + 1.2 MAC Sevo showed a reduction in astrocyte-mediated synaptic engulfment, which indicates reduced dendritic spine elimination by astrocytes.

As a specific phagocytic receptor, MEGF10 is critical in astrocyte-mediated synaptic engulfment. MEGF10-deficient astrocytes reduced the elimination of excitatory synapses, resulting in the accumulation of excessive but functionally impaired excitatory synapses, leading to cognitive impairment in mice [56]. Consistently, another study demonstrated that downregulation of MEGF10 in the hippocampus led to a decrease of engulfed presynaptic and postsynaptic markers, e.g., synaptophysin and PSD95, detected inside astrocytes [49]. As shown in our study, downregulation of MEGF10 in 3NTyrAβ + 1.2 MAC Sevo might reduce astrocyte-mediated synaptic engulfment. Due to the key role of MEGF10 in astrocyte-mediated synaptic engulfment, decreased MEGF10 expression reduced the phagocytic capacity of astrocytes [40], and might hence lower the clearance of excitatory synapses, thus partially counteracting the detrimental effects of 1.2 MAC sevoflurane on dendritic spines. This interaction may explain why 1.2 MAC sevoflurane did not enhance the elimination of DSD in the 3NTyrAβ preincubated hippocampal slices.

Interestingly, MEGF10-related astrocytic synapse elimination was not affected by the other conditions we tested, ie., Aβ_1-40_ + Sevo or Aβ_1-42_ + Sevo or AβpE3 + Sevo or 3NTyrAβ + 0.4 MAC Sevo. It is important to keep in mind that the experiments were conducted on acute hippocampal brain slices and differed from the clinical situation in which patients are confronted with factors such as hemodynamic instability, hypoxia, frequent occurrences of surgeries, and the potential impact of comorbidities such as cerebral vascular burden. Moreover, it is not yet clear whether the decrease in engulfment is beneficial or not, as the exact function of the eliminated spines is unclear. If they represent silent or malfunctioning synapses that should be eliminated, the downregulation of astrocyte-mediated synaptic engulfment will result in an elevation of non-functional or malfunctioning synapses and possibly produce physiological network distortion. Thus, in vivo experiments are pivotal to explore the effects of sevoflurane on dendritic spine remodeling and astrocyte-mediated synaptic phagocytosis. Furthermore, an important and crucial prerequisite is to detail whether the engulfed synapses are functional, silent or malfunctioning.

In conclusion, by investigating the interaction of sevoflurane and Aβ species on CA1 dendritic spine dynamics and MEGF10-related astrocytic synaptic elimination, our study showed a reduction of dendritic spines by either sevoflurane or Aβ alone or after co-application of Aβ with sevoflurane. Moreover, removal of sevoflurane partially reversed the spine loss in the presence of either Aβ species tested. In addition, 3NTyrAβ-downregulated MEGF10 and astrocytic-induced synaptic elimination impairment may compensate for the DSD reduction in the presence of Sevo. These results of the present study provide evidence for neuromorphological changes in AD patients receiving and recovering from sevoflurane anesthesia.

## 4. Materials and Methods

### 4.1. Animals

Adult male Thy1-eGFP and C57BL/6 mice (8–12 weeks old; Charles River, Munich, Germany) were used. All mice were kept and fed under standard conditions (12:12 h light/dark cycle, 22 ± 2 °C, 60% humidity) with free access to tap water and standard mouse food. Mice were randomly assigned to each group. All procedures were approved by the animal care committee (Technical University Munich, Munich, Germany) and were conducted in accordance with German law on animal experimentation. All efforts were made to minimize animal suffering and the number of animals used.

### 4.2. Preparation of Acute Hippocampal Brain Slices

Mice were anesthetized with isoflurane in an anesthesia chamber and decapitated after losing the righting reflex. The brain was quickly removed and placed into an ice-cold preparation ringer solution (125 mM NaCl, 2.5 mM KCl, 25 mM NaHCO_3_, 0.5 mM CaCl_2_, 6 mM MgCl_2_, 25 mM D-glucose, and 1.25 mM NaH_2_PO_4_). Sagittal hippocampal brain slices with a thickness of 400 μm were prepared using a vibratome (Leica VT1000s, Leica, Wetzlar, Germany). Acute hippocampal brain slices were then incubated in artificial cerebrospinal fluid (aCSF) (125 mM NaCl, 2.5 mM KCl, 25 mM NaHCO_3_, 2 mM CaCl_2_, 1 mM MgCl_2_, 25 mM D-glucose, and 1.25 mM NaH_2_PO_4_) oxygenated with carbogen (95% O_2_ and 5% CO_2_) at 35 °C for 30 min, followed by another 60 min at room temperature (21 °C to 24 °C).

### 4.3. Aβ Preparation and Incubation

Aβ (1 mg) was dissolved in 400 μL hexafluoro-2-propanol (HFIP), separated evenly into 20 microcentrifuge tubes, each containing 50 μL of solution. The tubes were placed in lyophilizer until white pellets formed at the bottom of the tubes, and the tubes were then tightly closed. We dissolved either Aβ isoform (50 µg) to a concentration of 100 µM by adding 111 mL dimethyl sulfoxide (DMSO) to one tube and then sonicated it in an ultrasonic bath to make Aβ dissolve completely. The whole procedure normally takes about 15 min [57]. Thereafter, Aβ was added to the beakers with CSF to reach a final concentration of Aβ to 50 nM.

### 4.4. Sevoflurane Application

We applied 0.4 MAC and 1.2 MAC of sevoflurane, corresponding to sedation and clinical anesthesia concentrations, respectively. As previous reported, the MAC of sevoflurane in C57BL/6 mice was about 3.25% [58]; therefore, in our study, the corresponding concentrations of sevoflurane in the gas phase for 0.4 MAC_rodent_ and 1.2 MAC_rodent_ were about 1.4% and 4% in sevoflurane vaporizer settings. Clinically sevoflurane was administered in the gas phase, therefore, to determine the aqueous concentrations of sevoflurane, samples from the recording chamber were taken and filled into airtight glass containers for gas chromatographic measurements. Previously, using acute mouse brain slices, we found that the aqueous MAC concentration of sevoflurane at room temperature (21 °C to 24 °C) is 0.38 mM [48]. Therefore, the corresponding aqueous concentrations of sevoflurane in our study were 0.15 mM (1.4%/0.4 MAC_rodent_) and 0.46 mM (4%/1.2 MAC_rodent_), respectively.

### 4.5. Experiment Schedule and Workflow

As shown in Figure 14 and Figure 15, after slicing, acute hippocampal brain slices were incubated in carbogen-saturated aCSF at 35 °C for 30 min, followed by another 60 min at room temperature (21 °C to 24 °C). The brain slices were then incubated with Aβ (at a final concentration of 50 nM) for 90 min. Next, sevoflurane at 0.4 or 1.2 MAC was applied for 90 min, followed by another 90 min of sevoflurane washout with carbogen. In the whole process, the acute hippocampal brain slices were always aerated with carbogen, and the control groups were gassed only with carbogen throughout the experiment.

### 4.6. Analysis of Dendritic Spines

Male Thy1-eGFP mice (8–12 weeks old) were used. After the different treatments, acute brain slices were fixed with 4% paraformaldehyde (PFA) (Merck, Darmstadt, Germany) over 48 h, then cryoprotected with 30% sucrose for another 3 days. Sections of 50 µm thickness were prepared using a cryotome. After 3 washes with 1 × PBS for 5 min, the sections were transferred to slides with coverslips. Images of CA1 pyramidal neuron dendritic spines were acquired by confocal microscope at 0.3 µm interval z-stacks with a 63×/1.40 NA oil-immersion objective. Only second-order apical dendritic segments were considered, 100–200 µm from the pyramidal soma (Figure 16). The images were taken with a Leica TCS SP8 X confocal microscope (Leica, Wetzlar, Germany) equipped with a “supercontinuum white light laser”.

The acquired dendritic images were zoomed in and analyzed with Leica Application Suite (LAS) X office 1.4.4 (Leica, Wetzlar, Germany), 20–50 µm in length, and 6–8 dendrites were analyzed per mouse. Dendritic spines were categorized as thin, stubby, and mushroom subtypes based on established criteria [59]. It is difficult to distinguish filopodia from long, thin spines; in our study, filopodia were classified as thin spines. Dendritic spines were classified as thin type if the diameter of the length was greater than that of the neck, and the diameters of the head and neck were similar. Dendritic spines were classified as mushroom type if the diameter of the head was much greater than that of the neck, and they were classified as stubby if the diameter of the neck was similar to the length of the spine. Dendritic spine density was expressed as the number of spines per 10 μm of dendrite. The categorization of dendritic spines was according to the previously established protocol in our lab [60].

### 4.7. Immunofluorescence and Colocalization Analysis

#### 4.7.1. Immunofluorescence Staining and Image Acquisition

Sections of 30 µm thickness were prepared with a cryotome and were then washed with 1 × PBS for 3 × 10 min and blocked with blocking solution (1 × PBS + 10% normal goat serum + 0.3% Triton-X-100) for 2 h at room temperature (21 °C to 24 °C). After this, the sections were incubated with primary antibodies (rabbit anti-PSD95 (1:400); mouse anti-GFAP (1:800)) on a shaker at 4 °C for two nights. The sections are then rinsed with 1 × PBS for 3 × 10 min and incubated with secondary antibodies (Alexa Fluor 488/647-conjugated goat anti-mouse/rabbit (1:500)) at room temperature (21 °C to 24 °C) for 2 h in the dark. After 3 × 10 min wash with 1 × PBS, sections were mounted with DAPI mounting medium.

High-resolution images of astrocytes from the stratum radiatum of the CA1 region of the hippocampus were selected for our study. We took 4–8 sections of images per animal with the Leica TCS SP8 confocal microscope. First, the hippocampal CA1 region was observed at low magnification (10/0.40 NA); we then switched to a 63×/1.40 NA oil immersion objective for single astrocyte acquisition. The z-stack was set according to the volume of the astrocyte. The lightening function of the confocal microscope was applied to deconvolve the images, which were then exported to the Imaris software 9.9.0 (Oxford, UK).

#### 4.7.2. Quantitative Analysis of Astrocyte-Mediated Synaptic Engulfment

The astrocytic engulfment of synapses was measured according to the previously established protocol in our lab [57]. The phagocytic index (PI) is the parameter % of ROI colocalized from the colocalization analysis of Imaris.

### 4.8. Western Blotting

#### 4.8.1. Sample Preparation

Male C57BL/6 mice (8–12 weeks old) were used for the Western blot analysis. After different treatments, the hippocampus was quickly dissected in ice-cold aCSF, then placed into ice-cold lysis buffer, ground with a pestle into homogenate, and afterward incubated for 30 min on ice. The homogenate was centrifuged at 12,000 rpm for 30 min at 4 °C and the supernatant was collected. Protein concentrations in the supernatant were determined by BIO-RAD’s DC protein assay. The total protein concentration of each sample was adjusted to 2 µg/µL by mixing with sample buffer and lysis buffer.

#### 4.8.2. Preparation of Sodium Dodecyl Sulfate-Polyacrylamide Gel Electrophoresis (SDS-PAGE)

The separation gels were prepared according to Table 1. N,N,N′,N′-tetramethyl ethylene diamine (TEMED) and ammonium persulfate (APS) were added to the solution. The separating gels were carefully sealed with Milli-Q water and left polymerized for 40 min at room temperature (21 °C to 24 °C). The stacking gel solution was then added on separation gels, and combs with 15 wells were placed into the stacking gel solution (ensuring that no air bubbles formed between the comb wells and the gel). The solution was polymerized at room temperature (21 °C to 24 °C) for 30 min. Freshly prepared SDS-PAGE is preferred for Western blotting.

#### 4.8.3. Electrophoresis and Transferring

After loading 20 μL of sample into each well of the stacking gel, the vertical electrophoresis setup was run at a constant voltage of 80 V for 30 min, and then switched to 120 V until the blue sample buffer ran out of SDS-PAGE. The separation gels were transferred to the transferring buffer (25 mM Tris and 192 mM glycine). Thereafter, the membrane was transferred at a constant voltage of 80 V for 1 h.

#### 4.8.4. Blocking and Antibody Incubation

The membrane was blocked with 10% Roti-Block (Roth, Karlsruhe, Germany) for 1 h on a 3D shaker at room temperature (21 °C to 24 °C). The membrane was then incubated with primary antibodies at 4 °C on a shaker overnight (Rabbit anti-MEGF10, Millipore, Burlington, MA, USA, 1:500; rabbit anti-GAPDH, Cell Signaling Technology, Danvers, MA, USA, 1:5000). On the next day, the membrane was washed with TBST for 3 × 10 min. Horseradish peroxidase-conjugated secondary antibody (1:10,000, anti-rabbit IgG; Cell Signaling Technology, Danvers, MA, USA) was applied at room temperature (21 °C to 24 °C) for 1 h. After that, the membrane was rinsed with TBST for 3 × 10 min, and the membrane was placed in Enhanced Chemiluminescence (ECL) detection reagent in a dark box with 30 s of shaking on a 3D shaker. The images were taken with ChemiDoc XRS + System 6.0.1 (Bio-Rad, Hercules, CA, USA) and Image-lab 6.0.1 (Bio-Rad, Hercules, CA, USA).

#### 4.8.5. Analysis of Western Blots

The ImageLab™ 6.0.1 (Bio-Rad, Hercules, CA, USA) was used for analyzing Western blot results by measuring the intensity of each band in increment images. The intensity was normalized with the stain-free blot image of the membranes taken under ultraviolet light. The band intensity of each lane was normalized to the total amount of protein in that lane, and all results were compared to the standard protein samples.

### 4.9. Statistical Analysis

Statistical analysis was performed using GraphPad Prism 8 software. For the comparisons of dendritic spine density (DSD, the average number of spines in a segment of dendrite 10 µm in length), Western blot, and astrocyte-mediated synaptic engulfment among three or more groups, one-way analysis of variance (ANOVA) followed by Dunnett’s multiple comparisons test was performed. For the comparisons of two groups, an unpaired *t*-test was performed. Data are reported as mean ± standard error of the mean (SEM). Statistical significance was defined at *p* < 0.05.

## Figures and Tables

**Figure 1 ijms-25-07393-f001:**
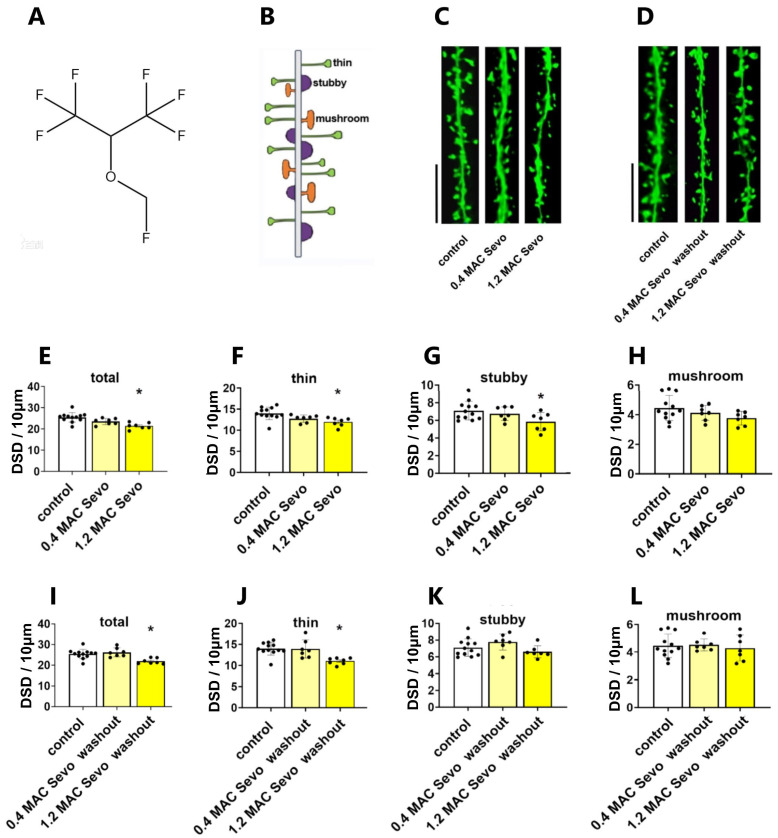
Sevoflurane decreases dendritic spine density (DSD). (**A**) Chemical structure of sevoflurane. (**B**) Schematic image of dendritic spines. (**C**,**D**) Representative apical dendritic segments of CA1 pyramidal neurons of all groups. Scale bars = 5 µm. (**E**) 0.4 MAC sevoflurane tended to reduce DSD (*p* = 0.0809), and 1.2 MAC sevoflurane significantly reduced DSD (*p* = 0.0003). (**F**) Both 0.4 and 1.2 MAC sevoflurane reduced thin spines (*p* = 0.0884 and 0.0042, respectively). (**G**) A downregulation of stubby spines only occurred in the 1.2 MAC sevoflurane group (*p* = 0.0276). (**H**) There was no change in mushroom spines under sevoflurane treatment. (**I**,**J**) After 90 min washout, the reduction of each subtype DSD produced by 0.4 MAC sevoflurane was completely reversed, while for 1.2 MAC sevoflurane, the decrease of total and thin spines persisted ((**I**), *p* = 0.0029 and (**J**), *p* = 0.0013). Stubby (**K**) and mushroom (**L**) spines did not change under Sevo washout. Sevo: sevoflurane. Data are shown as mean ± SEM. The number of points in (**E**–**L**) represents the number of animals. Every data point (black dots) represents the analysis of the slices of one individual animal. One-way ANOVA followed by Dunnett’s multiple comparisons test. * *p* < 0.05 represents statistical significance.

**Figure 2 ijms-25-07393-f002:**
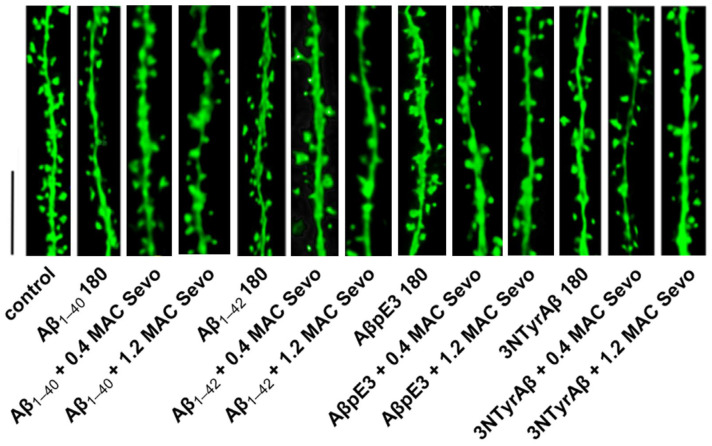
Dendritic spine images of Aβ + sevoflurane. Aβ 180: Aβ isoform incubated for 180 min. Sevo: sevoflurane. Scale bar = 5 µm.

**Figure 3 ijms-25-07393-f003:**
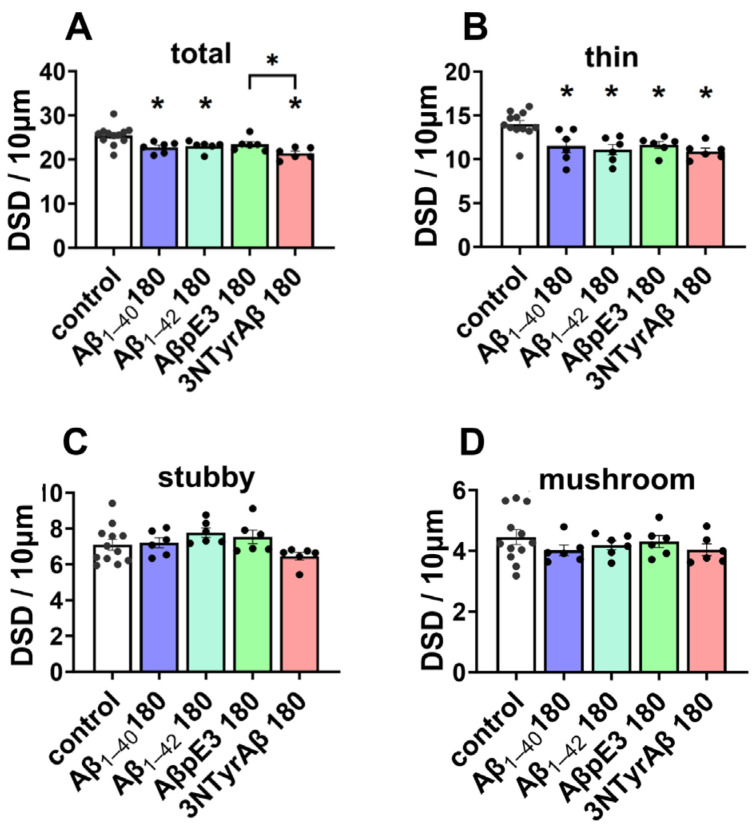
DSD under Aβ isoforms incubation for 180 min. (**A**) Aβ isoforms, including Aβ_1-40_, Aβ_1-42_, and 3NTyrAβ, but not AβpE3, reduced total DSD (*p* = 0.0132, 0.0305, and 0.0002, Dunnett’s test of one-way ANOVA). Moreover, 3NTyrAβ decreased DSD compared to AβpE3 (*p* = 0.0303). (**B**) Reduced thin spines under all conditions (*p* = 0.005, 0.009, 0.0074, and 0.0004, Dunnett’s test of one-way ANOVA). No change in stubby (**C**) and mushroom spines (**D**) in the presence of Aβ isoforms. Data are shown as mean ± SEM. Every data point (black dots) represents the analysis of the slices of one individual animal. * *p* < 0.05 represents statistical significance.

**Figure 4 ijms-25-07393-f004:**
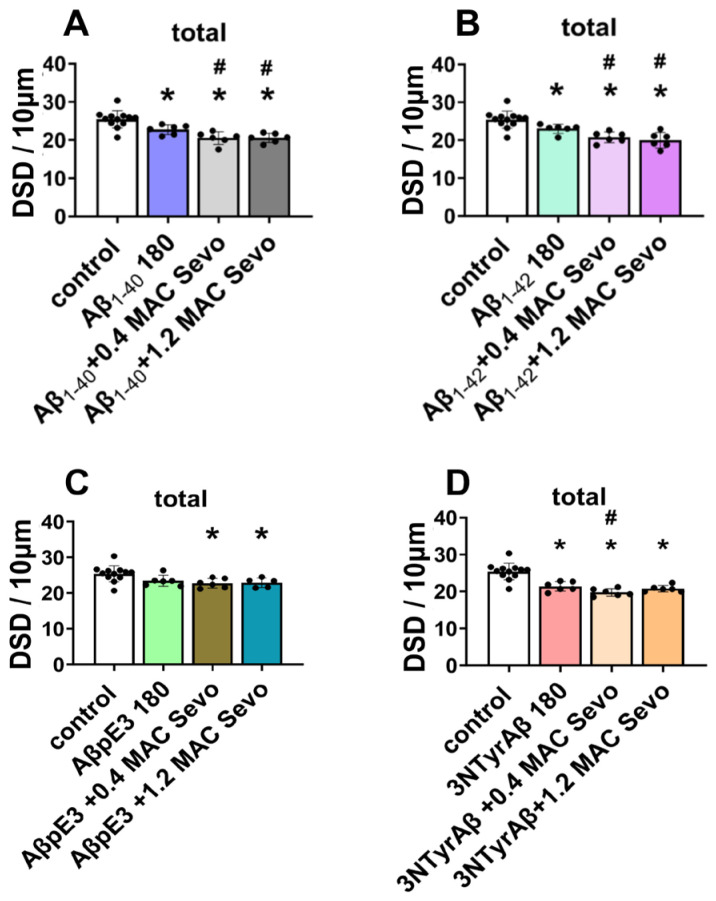
Total DSD in the presence of Aβ and sevoflurane. (**A**,**B**) DSD was reduced by Aβ_1-40_/Aβ_1-42_ (*p* = 0.0003/0.0003, in comparison with control), which was worsened by additional 0.4/1.2 MAC sevoflurane exposure (for 0.4 MAC Sevo: *p* = 0.0258/0.0488; for 1.2 MAC Sevo: *p* = 0.0304/0.0094). (**C**) Reduced DSD by AβpE3 + 0.4/1.2 MAC sevoflurane (*p* = 0.0187/0.0151). (**D**) NTyrAβ pretreatment reduced DSD (*p* = 0.0002), which was worsened only by 0.4 MAC sevoflurane (*p* = 0.0446). Aβ 180: Aβ incubated for 180 min. Sevo: sevoflurane. Data are shown as mean ± SEM. Every data point (black dots) represents the analysis of the slices of one individual animal. One-way ANOVA followed by Dunnett’s multiple comparisons test. *p* < 0.05 was used to represent statistical significance. *: significant difference from control; #: significant difference from specific Aβ.

**Figure 5 ijms-25-07393-f005:**
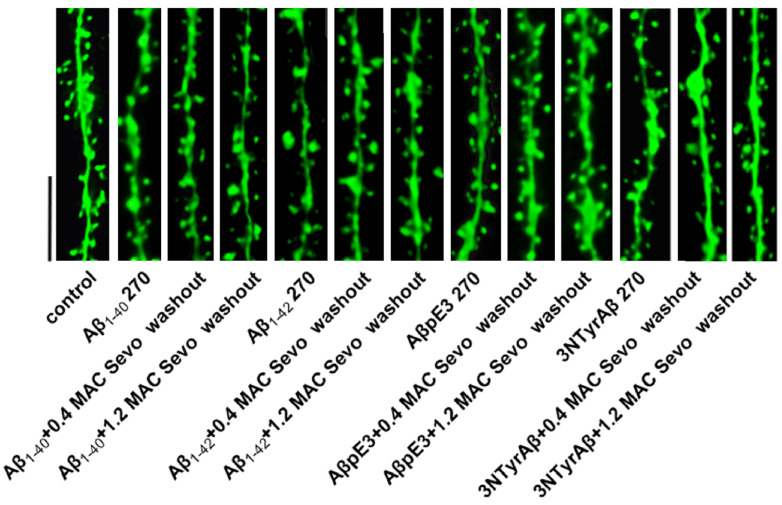
Dendritic spine images of Aβ-incubated hippocampal slices after the removal of sevoflurane. Aβ 270: Aβ isoform incubated for 270 min. Sevo: sevoflurane. Scale bar = 5 µm.

**Figure 6 ijms-25-07393-f006:**
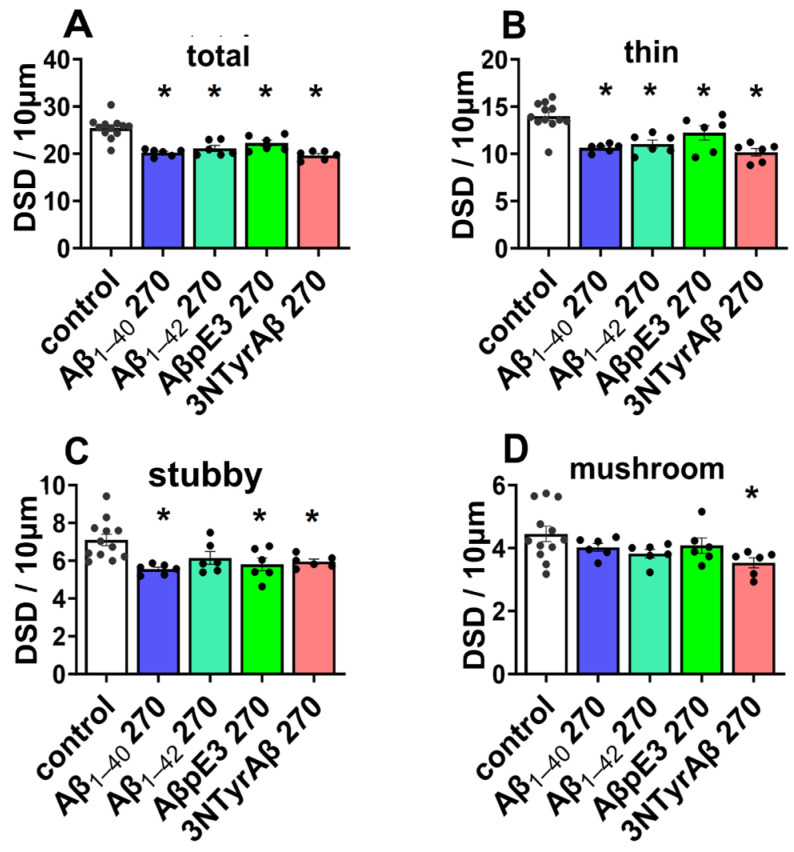
Dendritic spine density under Aβ species incubation for 270 min. All Aβ isoforms reduced total DSD (**A**, control vs. Aβ_1-40_ 270, Aβ_1-42_ 270, AβpE3 270, 3NTyAβ: *p* < 0.0001, <0.0001, =0.0028, <0.0001) and thin (**B**, control vs. Aβ_1-40_ 270, Aβ_1-42_ 270, AβpE3 270, 3NTyAβ: *p* < 0.0001, =0.0004, =0.452, <0.0001) DSD. (**C**) Reduced stubby spines under all conditions (control vs. Aβ_1-40_ 270, AβpE3 270, 3NTyAβ: *p* = 0.0020, =0.082, =0.0110, =0.0269) except Aβ_1-42_. (**D**) Mushroom spines were decreased only in the presence of 3NTyAβ (*p* = 0.0193). Sevo: sevoflurane. Data are shown as mean ± SEM. Every data point (black dots) represents the analysis of the slices of one individual animal. * *p* < 0.05 represents statistical significance.

**Figure 7 ijms-25-07393-f007:**
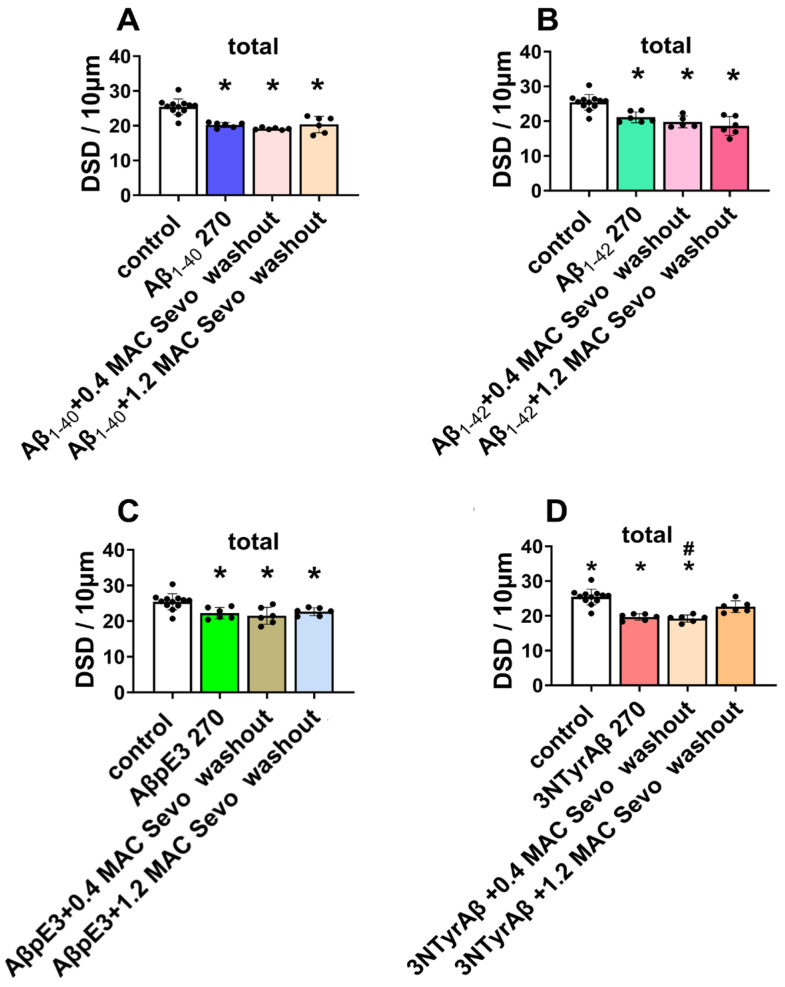
Total DSD in the presence of Aβ and sevoflurane washout. (**A**–**C**) Reduced DSD by Aβ was not changed after 0.4/1.2 MAC sevoflurane washout compared with control or Aβ270 min groups. However, 1.2 MAC sevoflurane + 90 min washout partially reversed DSD compared with 3NTyrAβ 270 min group (**D**, *p* = 0.0061). Aβ 270: Aβ incubation for 270 min. Data are shown as mean ± SEM. Every data point (black dots) represents the analysis of the slices of one individual animal. One-way ANOVA followed by Dunnett’s multiple comparisons test. *p* < 0.05 was used to represent statistical significance; *: significant difference from control, #: significant difference from specific Aβ.

**Figure 8 ijms-25-07393-f008:**
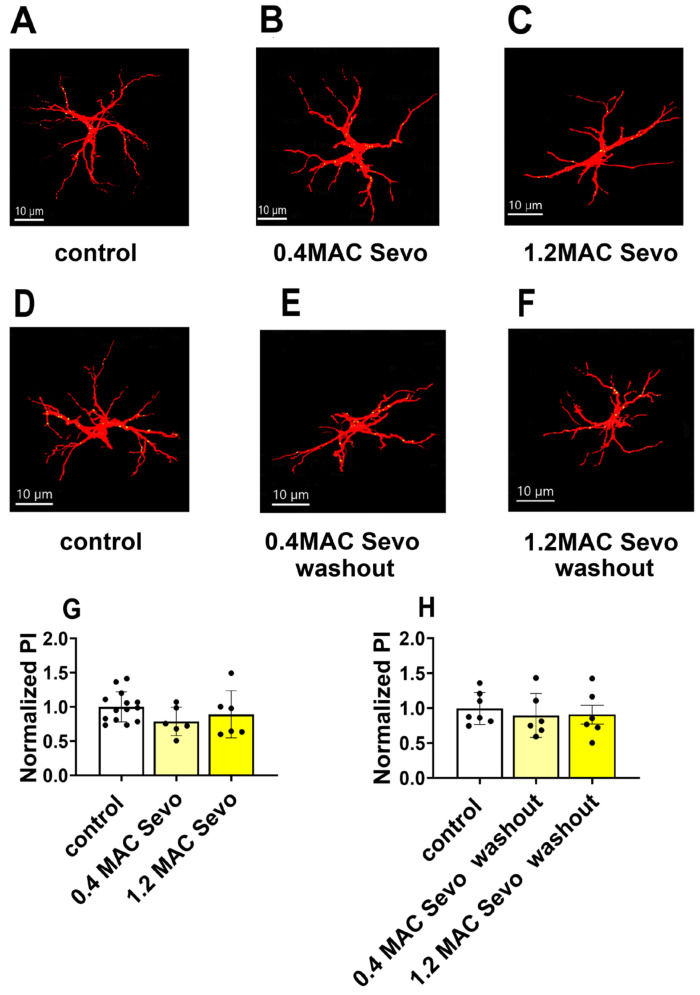
Neither 0.4 MAC nor 1.2 MAC sevoflurane had a significant effect on astrocyte-mediated synaptic engulfment. (**A**–**F**) Representative maximal projection images of engulfed PSD95 (shown in yellow) within GFAP-stained astrocytes (shown in red) in the control, 0.4 MAC Sevo, 1.2 MAC Sevo, 0.4 MAC Sevo washout, and 1.2 MAC Sevo washout groups, respectively. Imaris rendered individual astrocytes. The co-localization of PSD95 is shown in yellow within the astrocytes. (**G**,**H**) Compared to the control, neither 0.4 nor 1.2 MAC sevoflurane affected astrocyte-mediated synaptic engulfment. Sevo: sevoflurane. Every data point (black dots) represents the analysis of the slices of one individual animal. Data are shown as mean ± SEM. One-way ANOVA.

**Figure 9 ijms-25-07393-f009:**
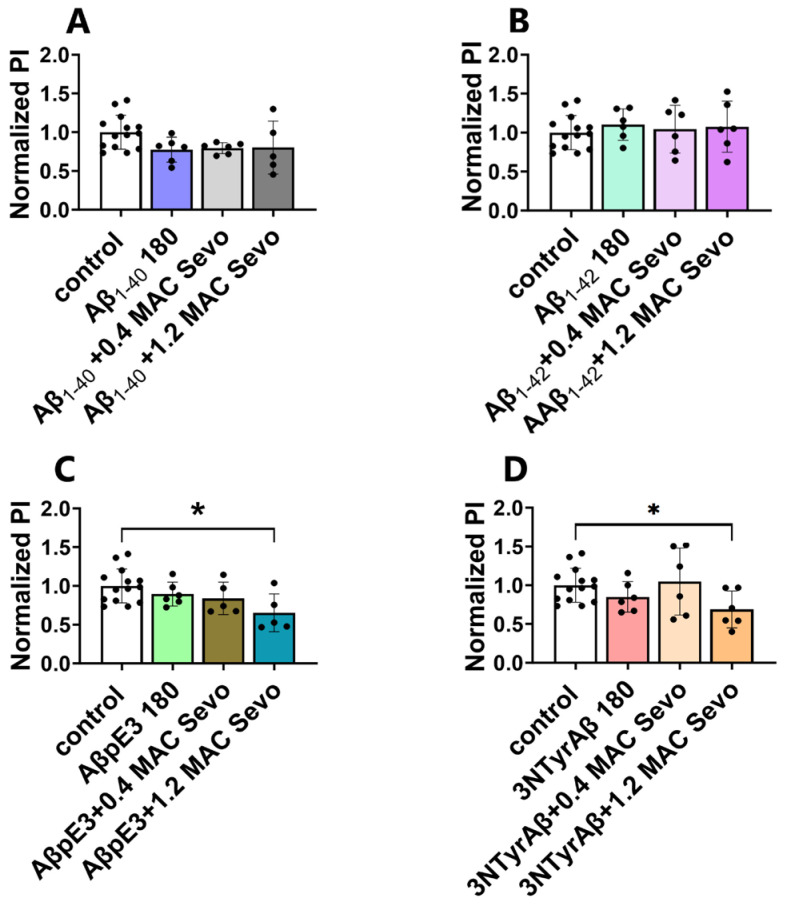
Co-application of 1.2 MAC sevoflurane with AβpE3/3NTyrAβ decreases astrocyte-mediated synaptic engulfment. (**A**,**B**) Neither Aβ_1-40_/Aβ_1-42_ nor Aβ_1-40_/Aβ_1-42_ + (0.4/1.2 MAC) Sevo affected the phagocytic index. (**C**,**D**) Both AβpE3 + 1.2 MAC Sevo and 3NTyrAβ + 1.2 MAC Sevo decreased the phagocytic index compared with the control group (*p* = 0.0088 and 0.0106, respectively). Aβ 180: incubation for 180 min. Sevo: sevoflurane. Data are shown as mean ± SEM. Every data point (black dots) represents the analysis of the slices of one individual animal. Unpaired *t*-test, * *p* < 0.05 represents statistical significance.

**Figure 10 ijms-25-07393-f010:**
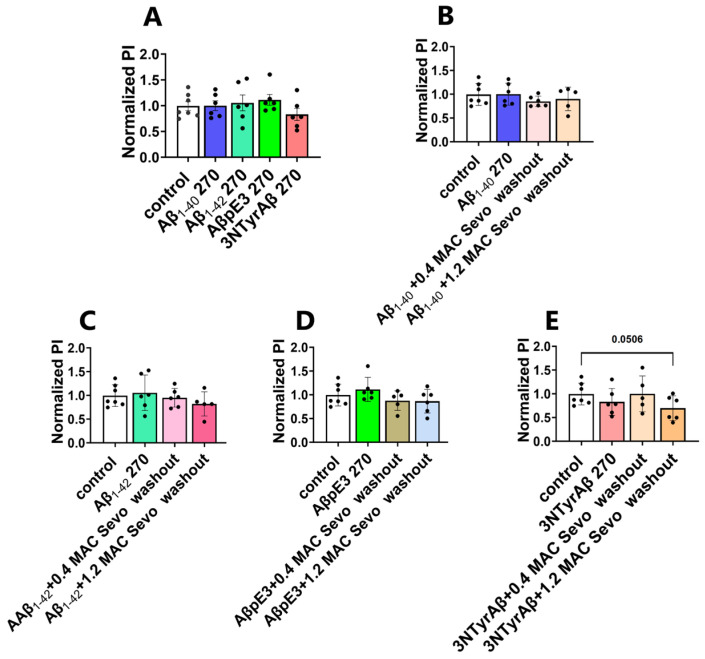
Removal of sevoflurane in 3NTyrAβ-treated slices decreases astrocyte-mediated synaptic phagocytosis. Among all conditions (**A**–**E**), the PI value was only reduced in the 3NTyrAβ + 1.2 MAC Sevo washout group (**E**). Aβ 270: incubation for 270 min. Sevo: sevoflurane. Data are shown as mean ± SEM. Every data point (black dots) represents the analysis of the slices of one individual animal. Unpaired *t*-test.

**Figure 11 ijms-25-07393-f011:**
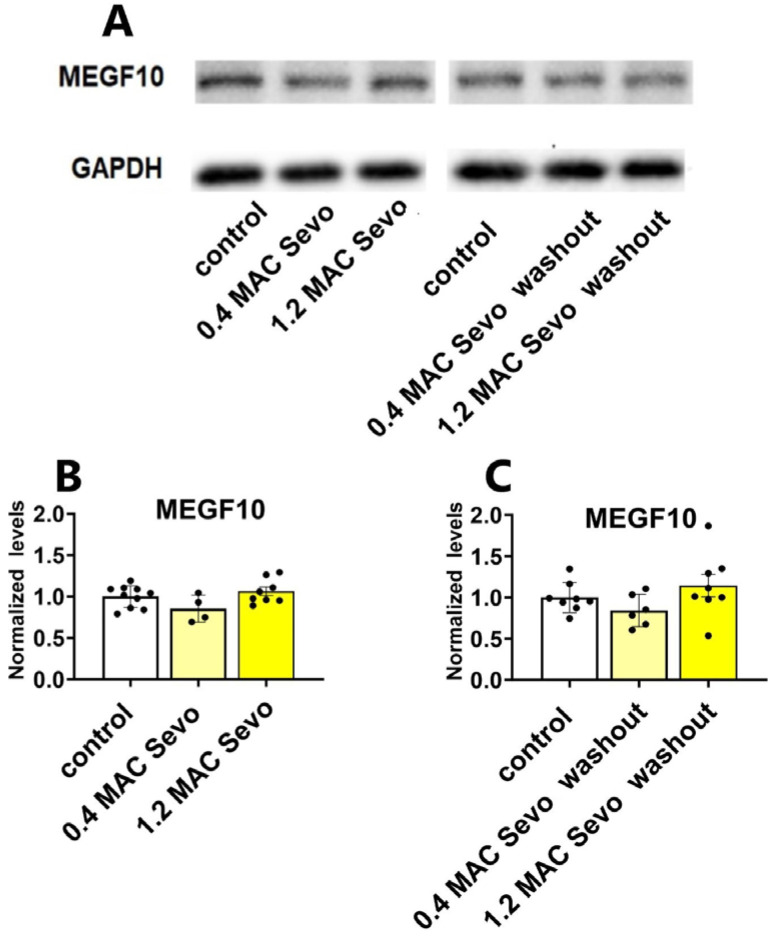
Neither 0.4 nor 1.2 MAC sevoflurane affects hippocampal MEGF10 levels. (**A**) Representative western blot bands of MEGF10 and GAPDH. (**B**,**C**) During exposure and after 90 min washout, 0.4/1.2 MAC sevoflurane did not affect the expressions of hippocampal MEGF10. Sevo: sevoflurane. Data are shown as mean ± SEM. Every data point (black dots) represents the analysis of the slices of one individual animal. One-way ANOVA followed by Dunnett’s multiple comparisons test.

**Figure 12 ijms-25-07393-f012:**
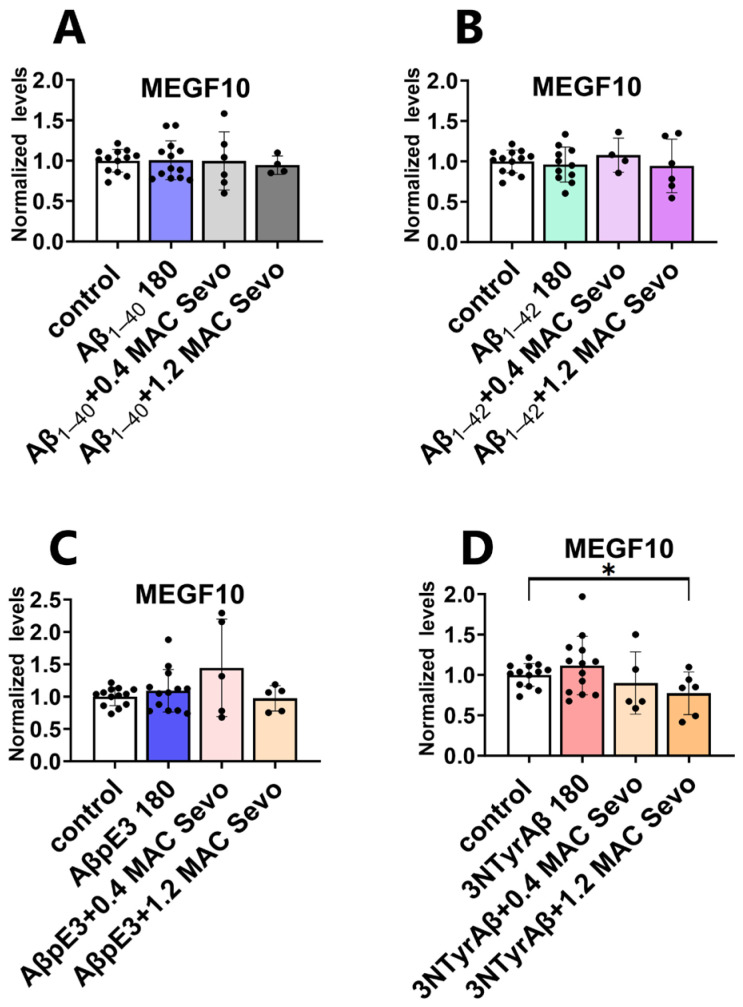
Co-application of 1.2 MAC sevoflurane with 3NTyrAβ downregulates MEGF10 expression. (**A**–**C**) Aβ_1-40_/Aβ_1-42_/3NTyrAβ alone or pretreatment of Aβ + 0.4/1.2 MAC sevoflurane did not change hippocampal MEGF10 levels. (**D**) MEGF10 was reduced in 3NTyrAβ + 1.2 MAC Sevo group compared to the control group (*p* = 0.024). Aβ 180: incubation for 180 min. Sevo: sevoflurane. Data are shown as mean ± SEM. Every data point (black dots) represents the analysis of the slices of one individual animal. Unpaired *t*-test; * *p* < 0.05 represents statistical significance.

**Figure 13 ijms-25-07393-f013:**
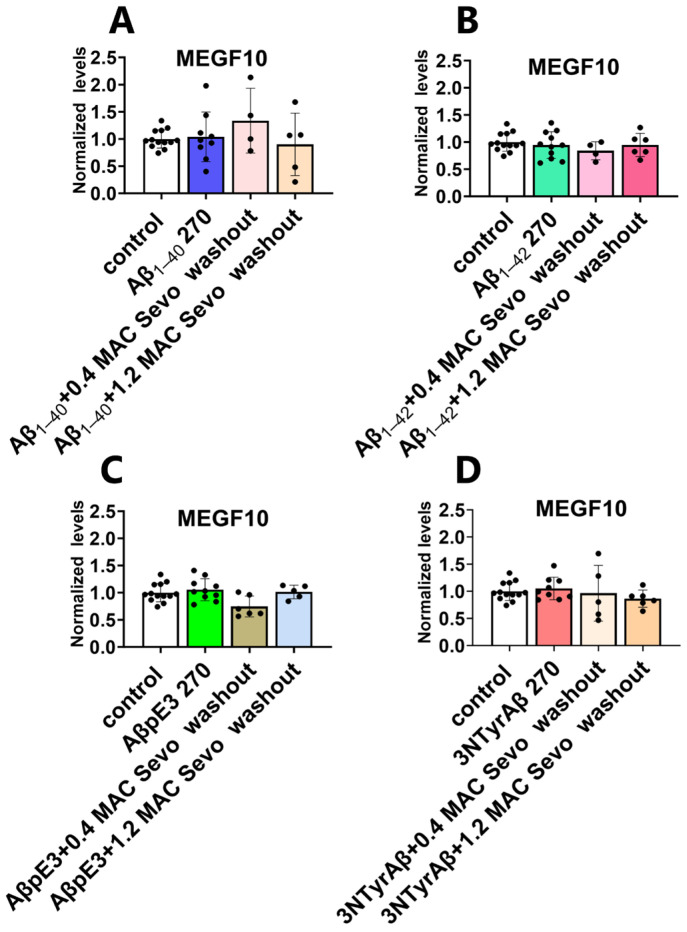
No alternation of MEGF10 levels under all conditions. (**A**–**D**) MEGF10 expressions were comparable among control, Aβ 270 and Aβ + Sevo washout groups. (**A**) No change of MEGF10 levels under Aβ_1-40_ 270/Aβ_1-40_ 270 + (0.4 MAC/1.2 MAC) sevoflurane washout. (**B**) Hippocampal MEGF10 expressions were not changed under Aβ_1-42_ 270/Aβ_1-42_ 270 + (0.4 MAC/1.2 MAC) sevoflurane washout groups. (**C**) AβpE3 270/AβpE3 270 + (0.4 MAC/1.2 MAC) sevoflurane washout did not alternate MEGF10 levels. (**D**) 3NTyrAβ 270/3NTyrAβ 270 + (0.4 MAC/1.2 MAC) sevoflurane washout had no effects on MEGF10 in the hippocampus. Aβ 270: incubation for 270 min. Sevo: sevoflurane. Data are shown as mean ± SEM. Every data point (black dots) represents the analysis of the slices of one individual animal. Unpaired *t*-test.

**Figure 14 ijms-25-07393-f014:**
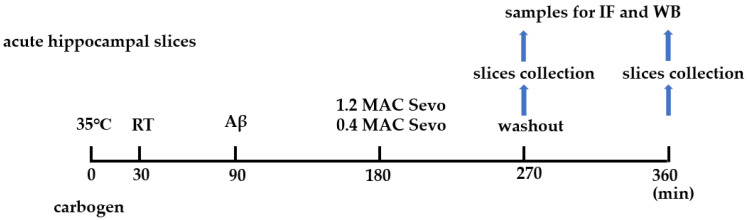
Schedule of Aβ incubation and sevoflurane exposure in ex vivo hippocampal brain slices. RT, room temperature; Sevo, sevoflurane; IF, immunofluorescence; WB, Western blot.

**Figure 15 ijms-25-07393-f015:**
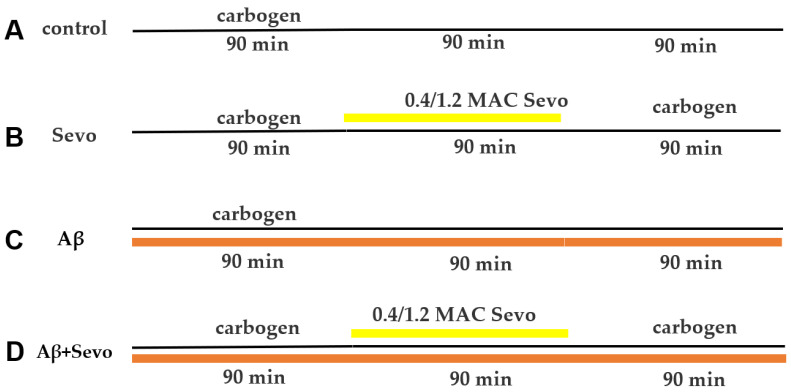
Timeline and workflow of Aβ application and sevoflurane exposure. (**A**) Control; the black line represents carbogen gassing. (**B**) Timeline of sevoflurane (Sevo) exposure and washing out. The black line represents carbogen gassing, and the yellow line represents Sevo gassing. (**C**) Timeline of Aβ incubation. The black line represents carbogen gassing, and the orange line represents Aβ incubation. (**D**) Timeline of Aβ incubation and sevoflurane exposure. The black line represents carbogen gassing, the yellow line represents sevoflurane gassing, and the orange line represents Aβ incubation.

**Figure 16 ijms-25-07393-f016:**
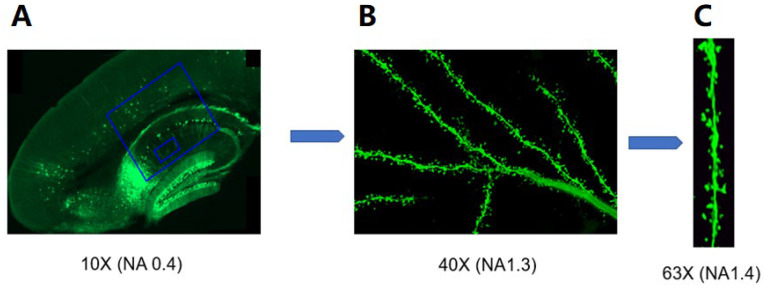
Overview of the process of dendritic spine acquisition. (**A**) Sagittal image of the hippocampus of a Thy1-eGFP mouse. (**B**) Apical dendrites of a pyramidal neuron. (**C**) Maximum intensity projection of a segment of dendrite.

**Table 1 ijms-25-07393-t001:** DSD under Aβ + sevoflurane. 
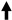
: significant increase, 

: significant decrease, 
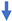
: a decrease with significance very close to 0.05; —: no change. Aβ 180 min: Aβ incubated for 180 min. Sevo: sevoflurane. Data are shown as mean ± SEM. One-way ANOVA followed with Dunnett’s multiple comparisons test. *p* < 0.05 was used to represent statistical significance.

DSD	Aβ_1-40_	Aβ_1-42_	AβpE3	3NTyrAβ	Compared to?
Thin	Stubby	Mushroom	Thin	Stubby	Mushroom	Thin	Stubby	Mushroom	Thin	Stubby	Mushroom
0.4 MAC Sevo			—			—		—	—			—	control
*p* = 0.0008	0.035	0.156	0.0006	0.040	0.768	0.012	0.154	0.610	<0.0001	0.014	0.082
—		—	—		—	—		—	—	—	—	Aβ180 min
0.542	0.041	0.809	0.976	0.007	0.987	0.714	0.040	0.974	0.960	0.115	0.677
1.2 MAC Sevo								—	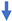				control
0.0002	0.019	0.014	0.027	0.043	0.044	0.035	0.160	0.051	0.005	0.006	0.009
—			—		—	—		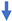	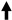		—	Aβ180 min
0.885	0.025	0.023	0.698	0.005	0.150	0.783	0.035	0.055	0.033	0.044	0.074

**Table 2 ijms-25-07393-t002:** DSD under Aβ + sevoflurane washout. 
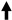
: significant increase, 

: significant decrease, 
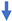
: a decrease with significance very close to 0.05; —: no change. Aβ 180 min: Aβ incubated for 180 min. Sevo: sevoflurane. Data are shown as mean ± SEM. One-way ANOVA followed by Dunnett’s multiple comparisons test. *p* < 0.05 represents statistical significance.

DSD	Aβ_1-40_	Aβ_1-42_	AβpE3	3NTyrAβ	Compared to?
Thin	Stubby	Mushroom	Thin	Stubby	Mushroom	Thin	Stubby	Mushroom	Thin	Stubby	Mushroom
0.4 MAC Sevo washout			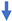			—			—			—	control
*p* < 0.0001	0.003	0.053	0.0005	0.025	0.105	0.037	0.003	0.146	<0.0001	0.002	0.062
—	—	—	—	—	—	—	—	—	—	—	—	Aβ270 min
0.618	0.994	0.347	0.629	0.672	0.887	0.995	0.637	0.758	1.000	0.185	0.755
1.2 MAC Sevo washout			—			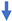		—		—			control
0.003	0.003	0.110	<0.0001	0.002	0.052	0.028	0.776	0.023	0.403	0.033	0.015
—	—	—	—	—	—	—	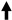	—	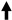	—	—	Aβ270 min
0.656	0.806	0.193	0.296	0.156	0.804	0.997	0.035	0.082	0.007	1.000	0.844

## Data Availability

Data supporting the findings of this study are available from the corresponding author upon reasonable request.

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
