# Peer review of "The Effects of Sevoflurane and Aβ Interaction on CA1 Dendritic Spine Dynamics and MEGF10-Related Astrocytic Synapse Engulfment"

_ijms, 2024, doi:10.3390/ijms25137393_

Round 1
Reviewer 1 Report
Comments and Suggestions for Authors
First of all I would like to thank the authors for the topic they have selected: anesthesic drugs impact in AD related biological changes. Studying potential modifiable risk factors for both clinical and biological progression of AD has a huge relevant.
However, I do have some concerns (minor) in relation to the manuscript that I would really appreciate the authors to answer:
- Abstract: Add Info about animal Model used and sample size for each category.
- Introduction: explain better clinical (syndromic diagnosis) and etiological diagnosis (biological) of AD, refering to the different stages of the diseases (preclinical, prodromal and Dementia stages) and to the sequential modificacions of ATN biomarkers. In turn, summarize in a table the most frequent cognitive and biological modifications in relation to AD continuum and other neurodegenerative etiologies described in relation to exposure to anesthesic drugs, specifying evidence at animal +/- human level,… Discuss how is frequent certain disocciation between what is observed in animal models and humans.
- Methods and results: Small sample size. I know that is frequent in this kind of studies but I am intrigued if you have considered to add female animals (gender disparities should also be considered) and different stages of AD (considered to add ptau patologhy for instance? Amyloid is an early phenomenon but ptau is more related situado clinical course and related specifically with synaptic damage. In turn, have you considered to assess posible excitability imbalance (glutamate related biomarkers for instance). And of course try to obtain information about cognition performace in relation to exposure. Also consider the fact to experiment with hemodynamic inestability, hypoxia… frequent in surgeries and the potential effect of comorbidities such as cerebral vascular burden.
- Discussion: add limitation section and another one to explore Research gaps and future Steps in the field. How would you suggest to continue with this study in humans? What are the clinical implications (potential) of the se results. Are really your results representative of AD patients or early preclinical stages (at risk). Compare results in relation to other etiologies of cognitive decline including vascular damage.
Reviewer 2 Report
Comments and Suggestions for Authors
1. Please provide the chemical structure and more biological/chemical properties of sevoflurane in Introduction section.
2. Why the authors tested AbetapE3 and 3NTyrAbeta? Do they have more neurotoxicity than regular Abeta (e.g., Abeta40, Abeta42)? How AbetapE3 and 3NTyrAbeta could affect the pathology of Alzheimer’s disease?
3. What are DSD and MAC? Please provide full name of those abbreviated terms in main text.
4. The authors tried to reveal the concentration-dependent effect of sevoflurane on DSD. However, In Figure 1, the authors have tested only 2 different concentrations of sevoflurane which is not enough to make a conclusion.
5. In general, the font size in the figure (e.g., axis title, legend) should be larger.
6. In line 190, the authors mentioned that “… was drawn Ab1-40/Ab1-42 + Sevo reduced stubby spines compared to …” however, the data is not shown in the manuscript. Is “Ab1-40/Ab1-42 + Sevo” means, either Ab1-40 + Sevo or Ab1-42 + Sevo?
7. The images of spines should be replaces with high-resolution images, especially, Figure 2A and 5A.
8. Why the authors tested the effect of Abeta+Sevo for 90, 180, or 270 min? Is there scientific reason?
9. Please keep in a same format for indicating “MEGF-10” in entire manuscript, either MEGF-10 or MEGF10.
10. In lines 315-317, the sentence “Here we found … unpaired t-test).” is not a proper conclusion. It is difficult to make this conclusion based on the data shown in Figure 12.
Round 2
Reviewer 2 Report
Comments and Suggestions for Authors
All of my concerns were cleared, however, the chemical structure of sevoflurane should be presented.
Author Response
Response to reviewer 2: Thank you for pointing this out. We have added the chemical structure of sevoflurane in Figure 1 and submitted the revised manuscript.